# Amplification of integrated microscopic motions of high-density [2]rotaxanes in mechanically interlocked networks

Xue Yang[1,2], Lin Cheng [1,2], Zhaoming Zhang[1,2], Jun Zhao[1], Ruixue Bai[1], Zhewen Guo[1], Wei Yu [1]✉ & Xuzhou Yan [1]✉

Integrating individual microscopic motion to perform tasks in macroscopic sale is common in living organisms. However, developing artificial materials in which molecular-level motions could be amplified to behave macroscopically is still challenging. Herein, we present a class of mechanically interlocked networks (MINs) carrying densely rotaxanated backbones as a model system to understand macroscopic mechanical properties stemmed from the integration and amplification of intramolecular motion of the embedded [2] rotaxane motifs. On the one hand, the motion of mechanical bonds introduces the original dangling chains into the network, and the synergy of numerous such microscopic motions leads to an expansion of entire network, imparting good stretchability and puncture resistance to the MINs. On the other hand, the dissociation of host−guest recognition and subsequent sliding motion represent a peculiar energy dissipation pathway, whose integration and amplification result in the bulk materials with favorable toughness and damping capacity. Thereinto, we develop a continuous stress-relaxation method to elucidate the microscopic motion of [2]rotaxane units, which contributes to the understanding of the relationship between cumulative microscopic motions and amplified macroscopic mechanical performance.

Mechanical motion is ubiquitous in living systems down to the level of tissues, cells, and beyond. And a plethora of essential life-sustaining functions strongly rely on the mechanical motions in a manner that individual microscopic motions couple to amplify to the macroscopic scales[1–3]. One of the most conspicuous examples is the skeletal muscle, in which actin filaments slide over myosin filaments on a length scale of 10–12 nm. Even so, their collective motions could give rise to a 1 $\mu$m contraction of the sarcomeres[4,5]. When the contractions occur in unison, a muscle is able to produce enough force to move the body[6]. Inspired by this, large numbers of artificial molecular muscle materials have been designed and prepared to perform useful tasks by taking advantage of the integrated and amplified molecular motions[7–15]. Nevertheless, it still remains a formidable challenge to produce macroscopic mechanical force along with the scale-up of motions at the molecular level in a single system, similar to what muscles do.

As one of the basic members of mechanically interlocked molecules (MIMs), rotaxanes are promising candidates for artificial molecular muscles on account of their abilities to simultaneously direct force and motion enabled by mechanical bonds[16–19]. The pioneering work of rotaxane-based molecular muscle was reported by the group of Sauvage, in which a metal-coordination-driven bistable [c2]daisy chain was demonstrated to be able to contract and stretch on demand with a nanometric displacement (∼1.8 nm)[20]. The contraction and extension of [c2]daisy chain-based molecular muscle were further scaled up over several orders of magnitude via a (supra)polymerization strategy[21–25]. For example, Giuseppone and coworkers achieved an amplification by four orders of magnitude−from nanometers to tens of

[1]School of Chemistry and Chemical Engineering, Frontiers Science Center for Transformative Molecules, Shanghai Jiao Tong University, Shanghai 200240, P. R. China. [2]These authors contributed equally: Xue Yang, Lin Cheng, Zhaoming Zhang. ✉e-mail: wyu@sjtu.edu.cn; xzyan@sjtu.edu.cn

micrometers–by coupling of thousands of [c2]daisy chains along a metallosupramolecular polymer chain[26]. However, these systems only mimicked a global change in scale in a muscle-like fashion without consideration of corresponding mechanical force. On the contrary, rotaxane cross-linked macromolecules oftentimes showcase good macroscopic mechanical performance due to the increased network homogeneity enabled by the mobile interlocked cross-links to release stress concentration in a cooperative manner[27–30]. For instance, Ito *et al.* developed a pulley-like rotaxane cross-linked system, namely the notable slide-ring materials (SRMs), which possess unusual stretchable, tough, and anti-tearing characteristics[31,32] (Fig. 1a). Takata *et al.* discovered a guiding strategy for strengthening and toughening polymer networks by utilizing small amounts of [2]rotaxane cross-linkers (≤0.5 mol%) with fine control over their mobile distance and mobility of the corresponding components[33,34] (Fig. 1b). It is worth noting that the favorable mechanical performance primarily stems from the pulley effect of the rotaxane motifs on conventional

polymeric chains, rather than their integration and amplification in structures[35–37]. Based on the above analyses, we envision that high-density integration of rotaxanes in a 3D network could provide a route to realize macroscopic response of mechanical force via scaling up the synchronized microscopic mechanical motions that muscle always does. Although it was targeted more than 40 years ago, coupling these two closely related aspects in a single system has been rarely explored to date.

Herein, we report a new class of mechanically interlocked networks (MINs) with densely rotaxanated backbones, which manifest themselves as an exceptional model system to showcase macroscopic mechanical properties originating from the amplification of integrated microscopic motion of the embedded [2]rotaxane motifs. In specific, a [2]rotaxane monomer bearing alkene units reacts with 3,6-dioxa-1,8-octandithiol (DODT) and pentaerythritol tetrakis(3-mercaptopropionate) (PETMP) through thiol−ene click chemistry to afford a network structure with high-density [2]rotaxane moieties (Fig. 1c). The network

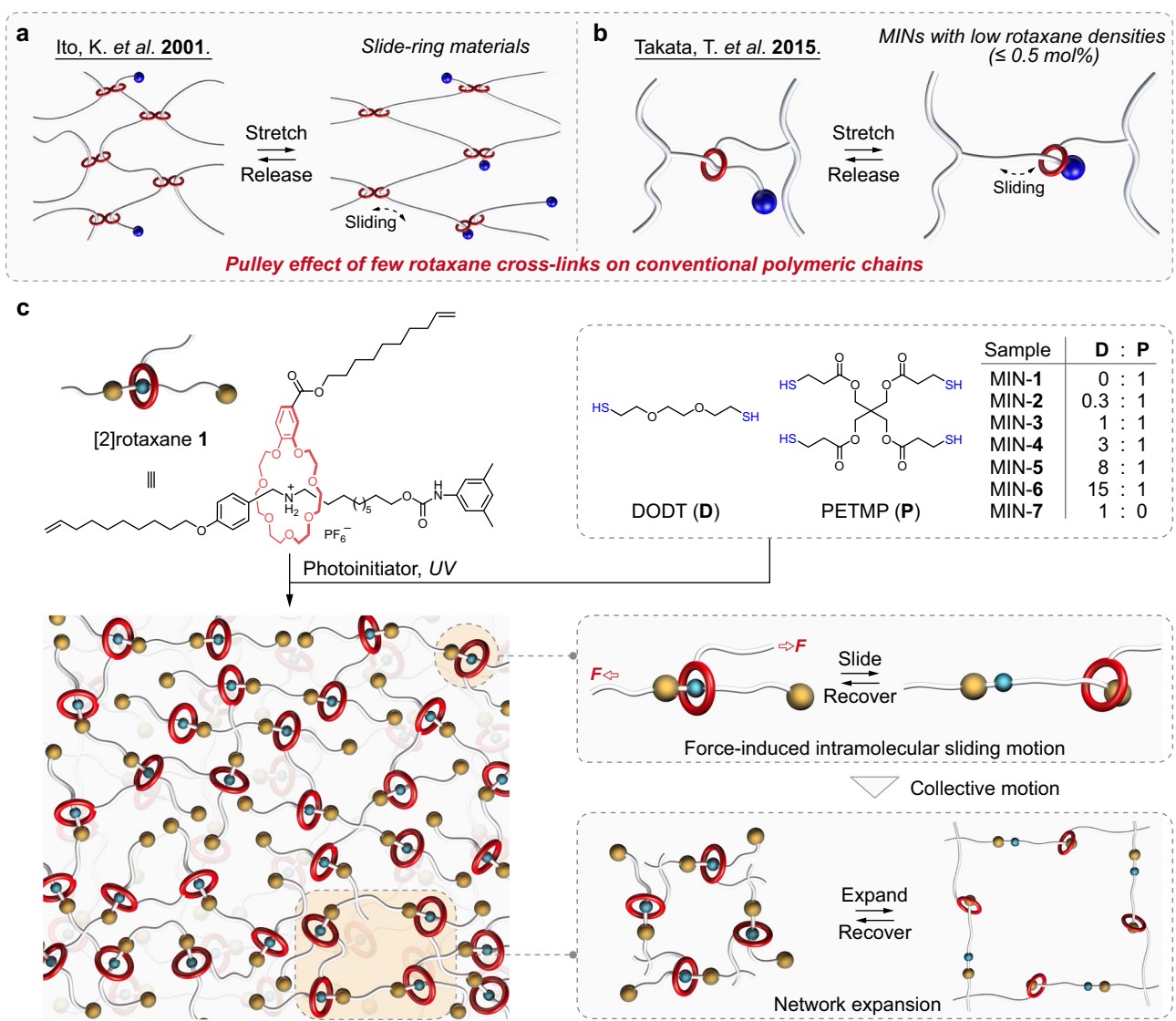

**Fig. 1 | Cartoon representations of the reported rotaxane cross-linked macromolecules and our MINs with high-density [2]rotaxane motifs. a** The sliding-ring materials reported by Ito et al.[31]. **b** The rotaxane-cross-linked polymeric networks reported by Takata et al.[33]. **c** Schematic representation of the formation of mechanically interlocked networks with densely rotaxanated backbones, in which the force-induced sliding motion of the [2]rotaxane unit and subsequent network expansion via the introduction of dangling chains as extended segments in a collective way are highlighted.

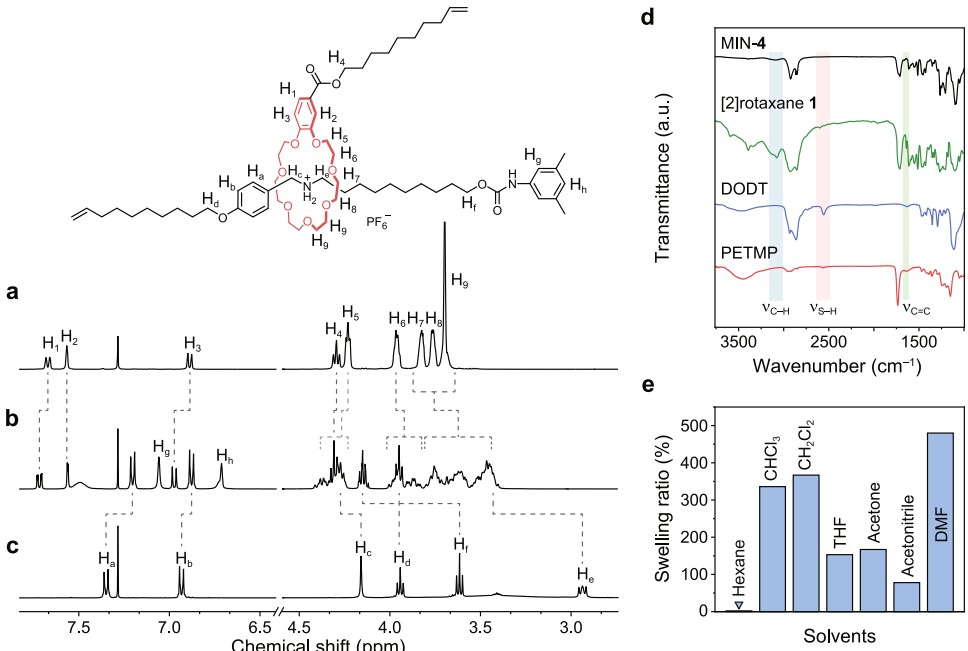

**Fig. 2 | Structural characterization of [2]rotaxane 1 and MINs. a–c** Partial $^1$H NMR spectra (CDCl$_3$, 400 MHz, 293 K) of (**a**) B21C7 wheel, (**b**) [2]rotaxane **1**, and (**c**) secondary ammonium salt axle. **d** FTIR spectra of MIN-**4**, [2]rotaxane **1**, DODT, and PETMP. **e** Swelling ratio histogram of MIN-**4** in different solvents.

structure is beneficial to transform the force applied on the bulk material to the microscopic mechanical bonds, giving rise to an intramolecular motion of the [2]rotaxane unit. The motion induces the original dangling short chain into the network structure, and the synergy of numerous such microscopic motions leads to an extension of the whole network, which endows the MINs with good stretchability (542%) and puncture resistance. Furthermore, the dissociation of host–guest recognition and subsequent sliding motion represent an efficient energy dissipation pathway, together with the integration and amplification effect, the bulk material exhibits favorable toughness (47.3 MJ/m$^3$) and damping capacity (95.4%). As a consequence, both the scale-up of mechanical motion and the resultant mechanical responsiveness are present in our MINs upon external force, reminiscent of the behaviors of muscles. In addition, the MINs also show a good recovery performance, which indicates that the mechanical bonds recover spontaneously after the intramolecular motion.

## Results

### Design, synthesis, and structural characterization

High density of mechanically interlocked units is the structural basis for the integration and amplification of microscopic mechanical motions in MINs. In our work, we employed the [2]rotaxane as a monomer to construct MINs rather than as a pure cross-linker of conventional polymers as reported[33], which could guarantee a high density of rotaxane moiety in the network. In specific, the [2]rotaxane monomer **1** was formed based on host–guest interaction between benzo-21-crown-7 (B21C7) wheel and secondary ammonium salt axle, of which one alkene unit was decorated each on the wheel and axle (Fig. 1c). Thanks to the designed [2]rotaxane, mechanically interlocked networks were constructed via photo-induced, radical-mediated thiol–ene click chemistry to transmit force on the mechanical bonds and ensure the integration and amplification effect. The variation in the feed molar ratios of DODT and PETMP enables facile modulation of the cross-linking densities of the target MINs. For this purpose, seven MINs named as MINs-**1**–**7** were prepared with the DODT/PETMP molar ratios of 0/1, 0.3/1, 1/1, 3/1, 8/1, 15/1, and 1/0, respectively. It is noteworthy that in all samples, the contents of the rotaxane units always maintain at a high level, because the amounts of rotaxane monomer in the reactions are weakly affected by the ratios of DODT/PETMP.

The $^1$H NMR spectra were firstly obtained to prove the preparation of the [2]rotaxane **1**. On the one hand, the aromatic protons H$_1$ and H$_3$ of B21C7 wheel moved downfield after the formation of [2]rotaxane **1** (Fig. 2a, b). Meanwhile, the ethyleneoxy protons H$_{5–9}$ on the wheel showed complexed signals because of the host–guest interactions. On the other hand, the aromatic protons H$_a$ and H$_b$ on the secondary ammonium salt motif showed a significant upfield shift while the methylene peaks H$_c$ and H$_e$ shifted downfield, which could be ascribed to the combined macrocyclic shielding effect and hydrogen bonds of [N–H···O] and [C–H···O] (Fig. 2b, c). In addition, the appearance of signals H$_g$ and H$_h$ validated the success of the stoppering process. These observations, together with the characterizations of $^{13}$C NMR and electrospray ionization mass spectrometry (ESI-MS) (Supplementary Figs. 9–11), clearly demonstrated the formation of the [2]rotaxane **1** and similar results were also observed on the [2]rotaxane **2** (Supplementary Figs. 13–15).

With the [2]rotaxanes in hand, we then performed the thiol–ene click polymerization to prepare the MINs and the control. The Fourier transform infrared spectroscopy (FTIR) measurements indicated that the polymerization reaction was complete (Fig. 2d and Supplementary Fig. 18). The characteristic peaks corresponding to C–H stretching vibration at 3076 cm$^{-1}$ and C=C stretching vibration at 1641 cm$^{-1}$ for the alkene unit of [2]rotaxane **1** and S–H stretching vibration at 2556 cm$^{-1}$ of the DODT and PETMP disappeared in MIN-**4**. These results implied the formation of the covalently cross-linked MINs, and the same phenomena were also observed in control (Supplementary Fig. 19). In addition, nano-mechanical measurement by atomic force microscopy (AFM) revealed that there was no obvious phase separation in the sample of MIN-**4**, indicating the formation of a homogeneous bulk material (Supplementary Fig. 20). We then conducted swelling experiments of MIN-**4** in seven solvents with different polarities to confirm the formed network structure. The MIN-**4** did not swell in nonpolar $n$-hexane, and even in good solvents of the other six, it only showed limited swelling ratios (<480%), indicative of the cross-linked nature (Fig. 2e and Supplementary Fig. 27).

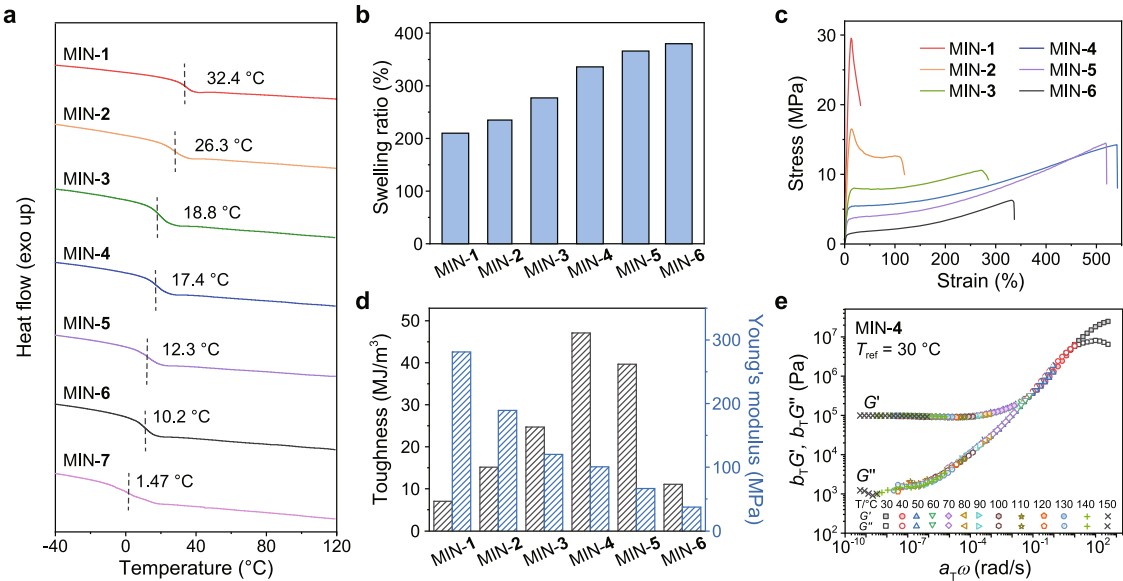

**Fig. 3 | Fundamental performance characterizations of MINs. a** DSC curves of MINs-**1**–**7** measured by the second heating scan from −40 to 120 °C with a rate of 10 °C/min. **b** Swelling ratios of MINs-**1**–**6** in CHCl₃. **c** Stress–strain curves of MINs-**1**–**6** with different molar ratios of DODT and PETMP. **d** Toughness and Young's moduli of MINs-**1**–**6** calculated from their corresponding stress–strain curves. **e** Master curves of MIN-**4** at a reference temperature of 30 °C.

## Fundamental performance characterizations of MINs

Network structure is closely related to the integration and amplification effect of MINs, and thus its influence on the properties of MINs was firstly investigated. Differential scanning calorimetry (DSC) was carried out to probe the glass transition temperatures ($T_g$) of MINs with different cross-linking densities. The results showed that the $T_g$ values increased from 1.47 to 32.4 °C with the increased content of PETMP cross-linker (Fig. 3a). Such a tendency was further confirmed by dynamic mechanical analysis (DMA) measurements (Supplementary Fig. 23). It is reasonable that denser networks are more likely to restrict the motion of the polymer segments, hence leading to higher $T_g$. Considering the fact that the motion of the mechanical bonds in the rubbery state of the material is more favorable than those in the glassy state, and thus the MINs with $T_g$ lower than room temperature are more suitable for the study of structure–property relationships.

Besides, the swelling behaviors of the cross-linked networks of MINs-**1**–**6** were also investigated. As shown in Fig. 3b, the swelling ratio from MIN-**1** to MIN-**6** exhibited an increasing tendency, which was consistent with the results of DSC, and further verified the cross-linking density of the samples. Then, we proceeded to study mechanical properties of the MINs with different cross-linking densities by tensile tests. The stress–strain curves of MINs-**1**–**6** were shown in Fig. 3c and the corresponding mechanical property profiles were summarized in Supplementary Table 2. Upon increasing cross-linking density, the Young's modulus grew significantly from 37.2 (MIN-**6**, DODT/PETMP = 15/1) to 281 MPa (MIN-**1**, DODT/PETMP = 0/1) (Fig. 3d). However, high cross-linking density undoubtedly sacrificed the tensile strain and toughness of the sample, as manifested by MIN-**1** whose maximum strain and toughness are 31.6% and 7.0 MJ/m³, respectively. With a medium cross-linking density, taking MIN-**4** for an example (DODT/PETMP = 3/1), its tensile strain and toughness reached maximum values of 542% and 47.2 MJ/m³, respectively, compared with the other MINs (Fig. 3c, d). Notably, because of the linear nature of MIN-**7** (DODT/PETMP = 1/0), it cannot form free-standing film for tensile test. These observations indicated that cross-liking density of the MINs had a significant effect on their mechanical properties. Due to the influence of cross-liking density, polymer networks generally exhibit a trade-off between swelling resistance and toughness; higher cross-linking density simultaneously leads to reduced swellability and toughness.

However, our MIN-**4** showed relatively low swellability (336%) but good toughness (47.2 MJ/m³), indicating the unique role of densely rotaxanated backbone in enhancing mechanical properties. Given a more balanced mechanical performance and also a proper $T_g$, MIN-**4** was chosen as a representative in the following study.

The network structure of MIN-**4** was clearly revealed by the master curves based on the time-temperature superposition (TTS) principle at a reference temperature of 30 °C. As illustrated in Fig. 3e, three regions on the master curves were shown in a wide frequency range of $10^4$ to $10^{-8}$ rad/s. The high frequency ($f > 10$) and the intermediate frequency ($0.03 < f < 10$) regions were assigned as the glassy and transition regions, respectively[38]. After the frequency lower than 0.03, a pronounced plateau with $G' > G''$ referred to as a rubbery plateau emerged, where the material exhibited a predominately elastic property, suggestive of a decent network structure.

## Macroscopic mechanical properties of high-density [2]rotaxane MIN

To better exhibit the property advantages of our MIN and understand its origin, a control sample was also designed, which consisted of [2] rotaxane **2** and the same thiol feeding ratio with MIN-**4**. The control could be regarded as a traditional covalent polymer network because the [2]rotaxane with two alkene groups on the ends of its axle, so that the B21C7 wheel cannot slide along the axle under applied force (Supplementary Fig. 29). These structural features were firstly confirmed by temperature sweep using DMA (Supplementary Fig. 24). When the temperature was around 40 °C, sharp decrease of storage modulus was observed for both MIN-**4** and control, which was caused by the glass transition of the networks. After that (with temperatures higher than 65 °C), the curve of control exhibited an obvious plateau, whereas the modulus for the curve of MIN-**4** continued to decrease slowly with the increase of temperatures. Such a phenomenon for MIN-**4** might be related to the motion of [2]rotaxane units after dissociation of host–guest recognition which could extend the network and thus lead to a reduced modulus.

As for the results of tensile tests for the two samples (Fig. 4a), MIN-**4** exhibited much better mechanical performances than those of the control in terms of stretchability (542 vs 268%), stiffness (Young's modulus: 101 vs 1.68 MPa), toughness (47.2 vs 1.70 MJ/m³), and

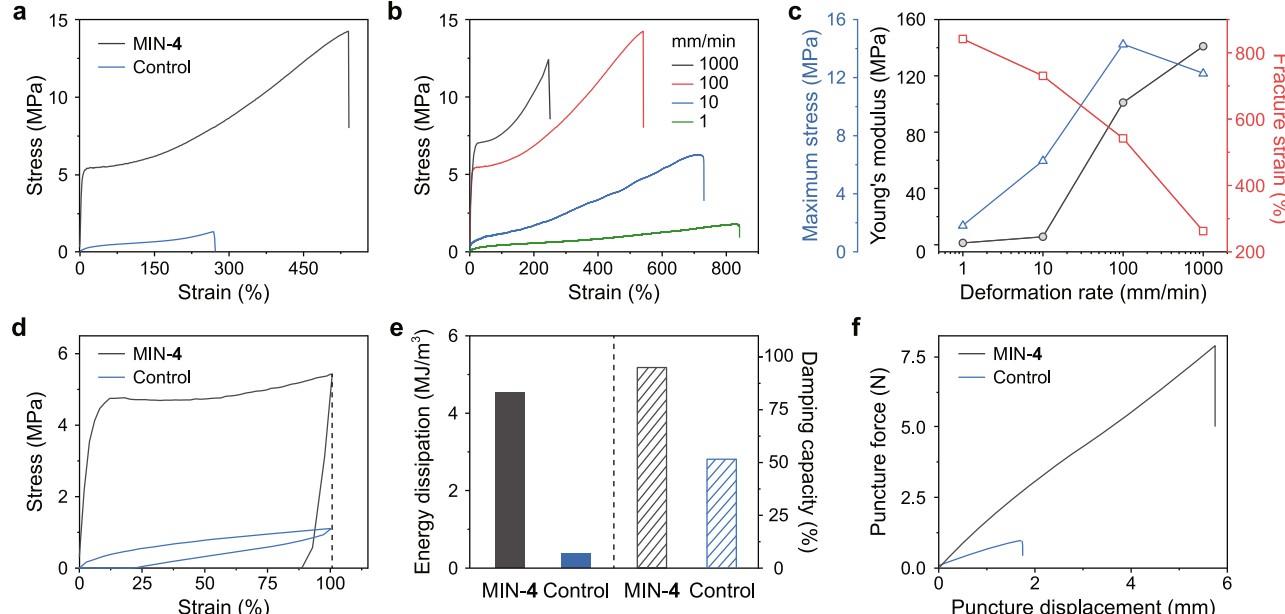

**Fig. 4 | Macroscopic mechanical properties of high-density [2]rotaxane MIN.**
**a** Stress−strain curves of MIN-**4** and control with a deformation rate of 100 mm/min. **b** Tensile test curves of MIN-**4** with different deformation rates from 1 to 1000 mm/min. **c** Deformation rate dependence of MIN-**4** on maximum stress, Young's modulus, and fracture strain. **d** Cyclic tensile test curves of MIN-**4** and control under applied strain of 100% with the deformation rate of 100 mm/min. **e** Energy dissipation and damping capacity of MIN-**4** and control calculated based on their cyclic tensile curves. **f** Puncture force−displacement curves of MIN-**4** and control. The puncture tests were conducted by puncturing the films with a sharp toothpick driven by an electronic universal testing machine.

strength (maximum stress: 14.3 vs 1.26 MPa), indicating the important role of slidable mechanical bonds in improving mechanical properties. Subsequently, the tensile tests with different deformation rates showed that the mechanical properties of MIN-**4** are stretching rate dependent (Fig. 4b). In detail, Young's modulus of MIN-**4** increased with the increase of deformation rate, and the same tendency was also observed for the maximum stress (Fig. 4c). Although the fracture strain kept reducing, the changes were relatively small compared to the other two parameters. It has been proved that the stretching rate-dependent mechanical behaviors are often observed for non-covalently cross-linked networks[39–42], and the noncovalent bond in our MIN could be attributed to the host−guest recognition.

Cyclic tensile tests for MIN-**4** and control were also performed (Fig. 4d). Under the same strain of 100%, the hysteresis area between the loading and unloading curves of MIN-**4** was about 12 times larger than that of control, as reflected by their energy dissipation values of 4.55 and 0.38 MJ/m³, respectively (Fig. 4e). Moreover, the ratio of energy dissipation to the incoming energy (damping capacity)[43] was calculated to be 95.4% for MIN-**4** and 50.7% for control (Fig. 4e). These results suggested that the dissociation of host−guest recognition and the consequent motion of mechanical bonds provided an effective pathway to dissipate energy.

Puncture resistance reflects the ability of a material to avoid breakage by a sharp object[44]. Corresponding tests for MIN-**4** and control were conducted by using a toothpick at a compression speed of 50 mm/min. For MIN-**4**, the maximal puncture force and displacement reached 7.85 N and 5.75 mm, respectively (Fig. 4f). The puncture energy of MIN-**4** was up to 23.5 mJ which was about 22 times higher than that of the control (1.09 mJ). The superior puncture resistance of MIN-**4** could also benefit from the intramolecular motion of the [2]rotaxane unit: when the B21C7 wheel slides along its axle, the chain length of the network strand between cross-links increases. As a result, the tension of the polymer chains can be equalized, which diminished the localization of the stress and thus showed good puncture resistance (Supplementary Fig. 31).

**Microscopic mechanical motion of [2]rotaxane moiety in MIN**
Now, the macroscopic mechanical features of MINs have been elucidated, but to correlate them with the integration and amplification effect, the mechanical motion of [2]rotaxane unit also needs to be revealed. For this purpose, a continuous multi-step stress relaxation (SR) approach on a strain-controlled rotational rheometer was adopted (Fig. 5a). The first step of stress relaxation (i.e., SR-1) was under a linear strain amplitude of 0.5% to obtain the initial relaxation behavior of MIN-**4**. Subsequently, the second step of stress relaxation (i.e., SR-2) was carried out at a defined constant strain amplitude ($\gamma_0$) ranging from 1 to 300% to trigger the mechanical motion of the [2]rotaxane moiety. We extracted the average plateau moduli ($G^0_{ave}$) from SR-2 curves in 60–100 s to evaluate the state of the sample under different strains. The normalized average plateau modulus ($G^0_N$) was calculated by the equation $G^0_N = G^0_{ave}/G_0$, where $G_0$ and $G^0_{ave}$ represent the average plateau moduli of the SR-1 and SR-2 stages, respectively. Finally, three consecutive stress relaxation tests (i.e., SR-3, 4, and 5) were performed under the strain amplitude of 0.5% to examine the response of the sample. It should be noted that the five consecutive stress relaxation curves represent a set of tests under each a defined strain.

As mentioned above, $G^0_N$ extracted from the stage of SR-2 could be employed to analyze the network behavior under different strains. There are prominent differences between the results of MIN-**4** and control as shown in Fig. 5b. Significant decrease of $G^0_N$ values for MIN-**4** started at smaller strains (<20%) than that for control (>30%), and the slope of its curve was also larger than that for control. In addition, even at small strains (<10%), the values of $G^0_N$ for MIN-**4** also declined slowly, which were well exhibited in Supplementary Fig. 32. It is well-known that polymeric networks under large strains would exhibit a decrease in modulus due to the network deformation. As for our MIN-**4**, the motion of mechanical bonds could also lead to the reduction of modulus, because sliding of B21C7 wheel on the axle is able to loosen the network[18]. Based on the results of MIN-**4**, we speculated that at small strains (<10%), the intramolecular motion of the [2]rotaxane unit in the network had already begun, but after the strain of 20%, the

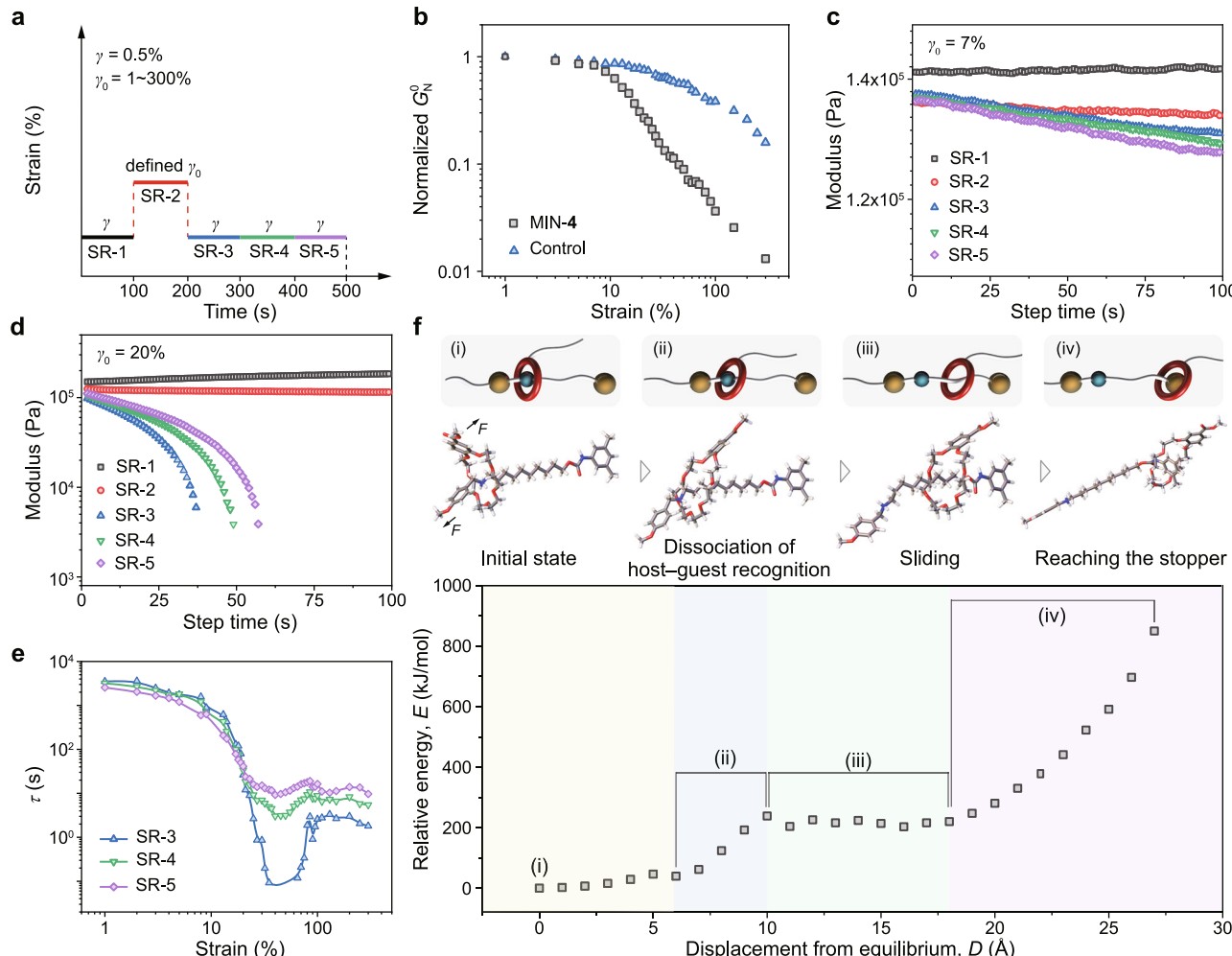

**Fig. 5 | Microscopic mechanical motion of [2]rotaxane moiety in MIN. a** The test schematic of consecutive stress relaxation for exploring the possible sliding behaviors of [2]rotaxane units in MINs by a rheometer. **b** The normalized average plateau modulus ($G^0_N$) of MIN-**4** and control as a function of shear strain. **c–d** Consecutive stress relaxation from SR-1 to SR-5 of MIN-**4** with the defined strains of (c) 7% and (d) 20%, respectively. **e** The relaxation time ($\tau$) of MIN-**4** as a function of shear strains based on SR-3, SR-4 and SR-5. **f** CoGEF potential as a function of stretched distance for a [2]rotaxane moiety (bottom), and corresponding schematic representation and CoGEF structures of the sliding behaviors (top).

mechanical motion and the deformation of the network structure took place simultaneously.

The speculation was verified by analyzing the whole process of SR-1−5. When the applied strain was set to be 7%, the stress relaxation results of the SR-1−5 were shown in Fig. 5c. Compared with SR-1, the modulus of SR-2 had a distinct decrease, which was consistent with the loosening of the network due to the intramolecular motion of [2] rotaxane moiety. During the processes of SR-3−5, the moduli declined continuously but slowly. Because SR-3 started immediately after the SR-2 without an interval, the decreased modulus indicated that after a relatively large-scale motion of mechanical bonds at SR-2, tiny mechanical motion continued to occur even under a small strain of 0.5% during the processes of SR-3−5. The same phenomenon was also observed for the experiment with the defined strain of 13% (Supplementary Fig. 33a). Such conclusion was further supported by the opposite results of the control in which a gradual increase of modulus was observed from SR-3 to SR-5 due to the recovery of the deformed network as analyzed below (Supplementary Fig. 34).

Furthermore, we increased the defined strain to 20% to probe the dynamic behaviors of [2]rotaxanes in MIN (Fig. 5d). Similarly, the modulus at SR-2 was also much lower than that of SR-1. However, different from the results in Fig. 5c, the stress relaxation curves of SR-3−5 displayed more notable relaxation behaviors, and the moduli

gradually increased from SR-3 to SR-5 at the same time. These were typical relaxation behaviors of deformation network. If a strain at the SR-2 stage gives rise to an elastic deformation of the network rather than just the motion of mechanical bonds, the deformed network would tend to recover to the original state at the SR-3 stage. The recovery process involves the retraction of the polymer chains, which would effectively relax the stress. And the elevated moduli from SR-3 to SR-5 reflected the reduced deformation of network strands in successive stress relaxations. This interpretation was verified by corresponding results for control which always presented gradual increased moduli in the procedures of SR-3 to SR-5 with a series of deformations (Supplementary Fig. 34). Furthermore, a more obvious phenomenon was exhibited by the test results with the defined strain of 30% (Supplementary Fig. 33b). Therefore, we can conclude that at small strains the motion of mechanical bonds took place, and larger strains (>20%) led to the mechanical motion accompanied by the elastic deformation of MIN.

To further clarify the dynamic behavior of the MIN upon different strains, the stress relaxation curves at the stages of SR-3−5 were fitted by the equation of $G(t) = G_0 \exp(-t/\tau)$, where $G_0$ is the elastic plateau modulus and $\tau$ is the macroscopic relaxation time of the network. As shown in Fig. 5e, there is a cross point at the strain of around 20%. When the strains were lower than 20%, the $\tau$ values were in the

following order: SR-5 < SR-4 < SR-3, indicating that SR-5 was able to relax stress faster than SR-3 and SR-4. Such a result was consistent with our above interpretation that under small strains, the motion of mechanical bond was dominant, and even under the strain of 0.5% during SR-3−5, the motion still slowly and continuously occurred. For the strains larger than 20%, a reverse trend was observed with the relaxation time in the following order: SR-5 > SR-4 > SR-3. As described above, the stage of SR-2 induced a large elastic deformation of the network whose recovery at stages of SR-3−5 caused notable stress relaxation, and most of deformation recovered at SR-3, thus leading to faster relaxation in this stage. Therefore, before the strain of 20%, the motion of mechanical bond was dominant, and after 20% strain, considerable elastic deformation of the network took place.

More information about the microscopic motion of mechanical bonds in MINs was provided by constrained geometries simulate external force (CoGEF) calculations[45] (Fig. 5f and Supplementary Fig. 35). When a [2]rotaxane moiety was under tension, the elongation profile could be divided into four regions[46]. The initial stage (0−5 Å) was relatively flat without significant change of the energy, and this stage was mainly related to the initial stretch of the chains in the molecular structure. Subsequently, an obvious energy increase (up to 238.5 kJ/mol) was observed in the second stage (6−10 Å), which could be ascribed to gradual dissociation of the host−guest interaction. The third stage (10−18 Å) exhibited a relatively stable energy profile, where the B21C7 wheel moved away from its ammonium station and slid along its axle, namely, a motion stage. Finally, an enthalpic regime (>18 Å) with a sharp increase in energy was observed. In this stage, the B21C7 wheel reached the site of the stopper, and then bond deformation occurred with the increase of strain. When the [2]rotaxane moiety sits in the fourth stage, further applied force would induce the elastic deformation of the network, as we analyzed above.

After proving the microscopic motion of mechanical bonds in MINs, the mechanical features of the MIN-**4** with densely rotaxanated backbone could be well understood based on the integration and amplification mechanism. From the perspective of structural design, two factors are mainly responsible for the mechanical behaviors of the MINs: one is the cumulative motion of dense mechanical bonds, and the other one is the cross-linked structures. Meanwhile, the role of the cross-linked structure on the mechanical properties could be evaluated by the control (Fig. 4), and thus the enhanced properties of the MIN-**4** compared with the control mainly stemmed from the integrated motion of dense [2]rotaxane units. For a single [2]rotaxane unit, the axial part between secondary ammonium salt site and stopper is initially presented as a dangling chain of the network, which becomes a part of the network after the sliding motion of B12C7 wheel along the axle upon stress. Due to the high-density [2]rotaxane units in the MINs, a large amount of such microscopic motion could be integrated and amplified into a distinct extension of the whole network, endowing the MIN-**4** with good performance in ductility and puncture resistance (Fig. 4a, f). Moreover, the dissociation of host−guest recognition combining with subsequent sliding of B21C7 wheel provide an effective pathway to dissipate energy. Based on the contribution of numerous [2]rotaxane units, our MIN-**4** is capable of dissipating the applied energy efficiently, showing markedly improved toughness. In a word, through integrating and amplifying microscopic motion of [2]rotaxane units, the MINs achieve the scale change of the network structure and macroscopic response of mechanical force simultaneously, which is reminiscent of the working mechanism of muscles.

Such a mechanism is largely different from that of the well-known rotaxane-crosslinked polymers[33,34]. From the perspective of structure, the content of rotaxane units in the system is relatively low, and the main part of the network is still the conventional polymeric backbone. In this case, the rotaxane acts as a crucial functional unit which could modulate the stress distribution on conventional polymer chains. In a word, a small amount of rotaxane units is able to efficiently tune and improve the mechanical properties of conventional polymers, which represents a very smart and efficient design. For our MINs with densely rotaxanated backbones, rotaxane is the monomeric unit and main constitution of the network. The mechanical properties of MINs fully originate from integrating and amplifying the motion of rotaxanes. Therefore, the macroscopic mechanical properties of MINs are the direct reflection of microscopic motion of [2]rotaxanes. To sum up, rotaxane-crosslinked polymers and our MINs with densely rotaxanted backbones are two different systems, and each of them has its own working mechanisms and merits. For example, the preparation of rotaxane-crosslinked polymers is facile and they hold great potential in practical applications. However, the integration and amplification of stimuli-induced microscopic motion of rotaxanes represent a unique mechanism in polymer physics which is also worth in-depth study from the fundamental point of view. Moreover, the integration and amplification of microscopic motion in MINs would result in superior actuation behaviors on which we will put more efforts in our future work.

### Recovery properties of high-density [2]rotaxane MIN

The force induced motion of [2]rotaxane moiety has been elucidated, then we attempt to explore the recovery property of the mechanical bond after unloading. To this end, we developed a combined rheological method to evaluate the recovery behaviors of MINs under large strains[47] (Fig. 6a). Firstly, a frequency sweep in linear viscoelastic region was carried out to characterize the initial state of our MINs. Then, a flow ramp was performed from 0 to a defined shear rate in each set of testing, where the range of the defined shear rate ($\dot{\gamma}$) increased from 0.02 to 0.64 s$^{-1}$. Accordingly, the range of shear strain ($\gamma_0$) grew from 1 to 1024% by the conversion of shear rate. This step was equivalent to apply different strains or stresses on MINs in the shear flow field. After that, a stress relaxation was conducted under maximum reverse strain amplitude ($-\gamma_0$) to eliminate the applied strain or stress in each flow ramp step. Finally, a frequency sweep like the initial step was conducted to verify the degree of recovery for MINs. As shown in Fig. 6b, even the shear strain $\gamma_0$ reached up to 1024%, the frequency sweep curves at the first and last stages overlapped well with each other. And the measurements under other shear strains also exhibited similar phenomena (Supplementary Fig. 37). Further, cyclic temperature ramp tests of MIN-**4** were also conducted (Fig. 6c). Similar to the aforesaid results, the temperature ramp curves almost overlapped completely. These results indicated that the B21C7 wheel pulled away from the secondary ammonium salt site could move back quickly in the rheological tests.

To further study the recovery performance of MIN-**4**, the creep and recovery experiments were carried out at room temperature (Fig. 6d). Upon loading, creep strain gradually increased, and the strain reached ∼75% after 45 min. When the applied stress was released, the strain recovered largely with the residual strain lower than 5% within 55 min. In addition, the recovery property at room temperature was further investigated by cyclic tensile tests (Fig. 6e). The cyclic tensile curve of MIN-**4** at a strain of 200% showed an obvious hysteresis loop, indicating efficient energy dissipation caused by the sliding motion of [2]rotaxanes during stretching. Then the second cycle without relaxing showed a small hysteresis loop, which was possibly because partial temporarily dissociated host−guest interactions did not fully recover in the time scale of an individual cycle. As the extension of rest times, the hysteresis area increased gradually, indicating the gradual recovery of the host−guest recognition and the polymer network. After about 30 min, the curve almost completely overlapped with the first cycle and the recovery ratio finally reached up to 98 % (Supplementary Fig. 41). The results of creep and cyclic tensile tests implied that even worked at room temperature, the force-induced sliding motions are reversible.

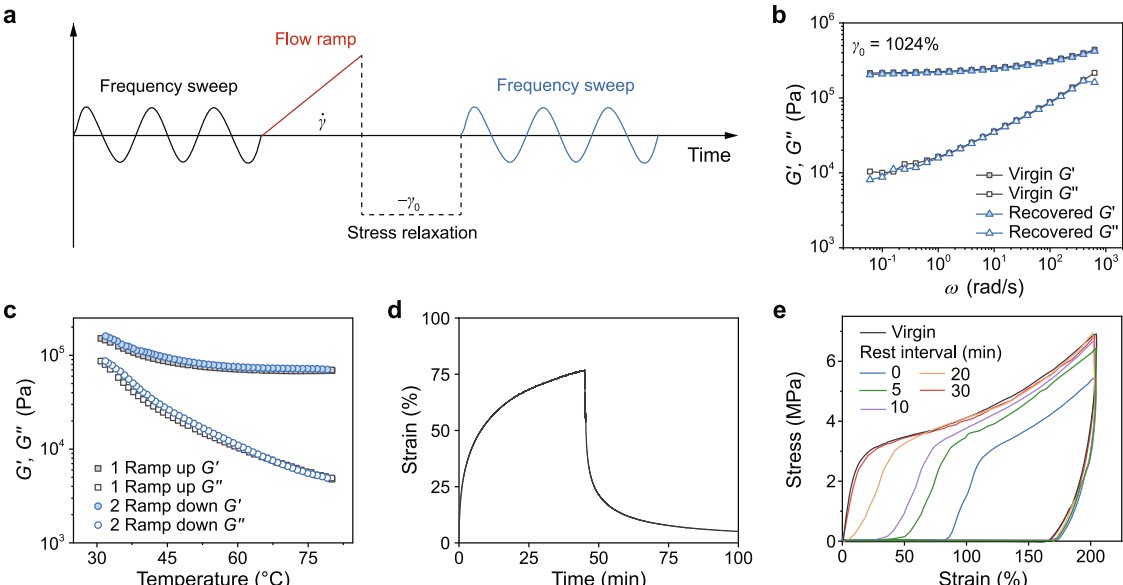

**Fig. 6 | Recovery properties of high-density [2]rotaxane MIN. a** The test protocol of combined rheological method for the characterization of recovery of MIN-**4**. **b** The virgin and recovered $G'$ and $G''$ of MIN-**4** versus frequency ($\omega$) under a shear strain of 1024% obtained by the above-mentioned method. **c** Cyclic temperature ramp curves of MIN-**4** ranging from 30 to 80 °C with a rate of 5 °C/min. **d** Creep and recovery curve of MIN-**4** at a constant stress of 0.2 MPa. **e** Cyclic tensile test curves of MIN-**4** loaded at a strain of 200% with rest intervals from 0 to 30 min, respectively.

## Discussion

In summary, we have designed a series of MINs-**1**–**7** with densely rotaxanated backbones, which were prepared by thiol–ene click chemistry between an alkene-decorated [2]rotaxane monomer and comonomers of DODT and PETMP. By controlling the ratio of DODT/PETMP, the cross-linking density of the MINs was effectively tuned, which was crucial to the mechanical properties of MINs. Compared with control whose mechanical bonds cannot move upon stress, MIN-**4** exhibited improved mechanical properties in stiffness (Young's modulus = 101 MPa), strength (maximum stress = 14.3 MPa), stretchability (542%), and toughness (47.2 MJ/m$^3$), and also had much better performance in energy dissipation and puncture resistance. The macroscopic mechanical features of MINs were demonstrated to originate from the integration and amplification of microscopic motion of the mechanical bonds in high-density [2]rotaxane network. Assisted by rheology technique and CoGEF simulation, the microscopic mechanical motion of mechanical bonds was revealed. Moreover, studies on the recovery property of MINs implied that the B21C7 wheel pulled away from secondary ammonium salt site was able to move back after unloading. Our work demonstrates that the integration and amplification of high-density mechanical bonds represent an effective and powerful approach to develop high-performance mechanically interlocked materials, which holds great promise for potential applications in stretchable electronics, anti-scratch coatings, microactuators, and so on.

## Data availability

The authors declare that the data supporting the findings of this study are available within the paper and its supplementary information files, or the data are available from the corresponding authors on request. Source data are provided with this paper.

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

## Acknowledgements

X.Y. acknowledges the financial support of the NSFC/China (22122105, 22071152, and 21901161) and the Natural Science Foundation of Shanghai (20ZR1429200). Z.Z. acknowledges the financial support of the NSFC/China (22101175). W.Y acknowledges the financial support of the NSFC/China (51625303). This research is supported by the Starry Night Science Fund of Zhejiang University Shanghai Institute for Advanced Study, Grant No. SN-ZJU-SIAS-006.

## Author contributions

X.Yan and W.Y. supervised this research. X.Yan and X.Yang conceived the project. X.Yang carried out the synthesis and some characterizations of the materials. R.B. synthesized some chemical intermediates. L.C. carried out the rheological tests under the supervision of W.Y. Z.G. performed the CoGEF calculations. Z.Z., L.C. and J.Z. helped with the manuscript preparation. The manuscript was written by X.Yang, Z.Z., W.Y., and X.Yan with contributions from all the coauthors.

## Competing interests

The authors declare no competing interests.
