## [Peer Review File · Nature Communications]

Amplification of integrated microscopic motions of high-density [2]rotaxanes in mechanically interlocked networksReviewers' Comments:

Reviewer #1:

Remarks to the Author:

This revised manuscript can be published as it is now.

Reviewer #2:

Remarks to the Author:

In the revised manuscript by Yan et al, they have collected the data for the reference polymer and addressed the temperature dependence of the MIMs. They also provided the clear evidence on the ionic interactions. I believe that they have made a better version according to the referees' comments. However, there is room for further improvement and I list my suggestions below for their references. I would like to support the publication on Nature Communication after minor revision.

1. Despite the authors have removed the discussion on the comparison between MIM-4 and other reported RCPs in the SI, they should also remove the related discussion in the main text and Figure 3e. Such discussion and presentation provide a clear misleading on the fact that the swelling ratio is highly dependent on the polymer backbone. Instead of putting other PCRs, the author should focus on MIN-1 to MIN-7 here if they want to highlight the mechanical performance.

2. I would suggest authors to put the temperature dependence results of the MIM-4 and the control data in the main figures and have a proper discussion, this is an important aspect for the MIMs based on H-G interactions.

3. I agree that the authors are trying to elucidate the integration and amplification effects on the mechanical bond in such polymeric systems, but the authors should indicate / emphasise what are the real merits arising from the densely rotaxanated backbones instead of putting a small amount of rotaxane in a cross-linked polymer.

4. The authors mentioned that the preparation of the deactivated rotaxane reference was failed. What are the conditions the authors used? Could authors try strong bases (such as DBU, NaH) to deprotonate the crown-ether ammonium salt instead of acetylation.

Reviewer #3:

Remarks to the Author:

The authors prepared [2]rotaxane-based polymeric materials with high mechanical properties. The authors experiments and discussions to evaluate amplification effect in molecular level were interesting and well-designed. However, we do not recommend accepting this manuscript in Nature Communication because of the reasons in below. It should be suitable more specified journal as the author studied well about their materials from various aspects.

1. Materials design

The authors' molecular design was similar to a molecular design reported by Takata et al. [Angew. Chem. Int. Ed., 2019, 58, 2765-2768, DOI:10.1002/anie.201813439] and the recently published molecular design by authors [J. Am. Chem. Soc. 2022, 144, 872-882, DOI: 10.1021/jacs.1c10427].

2. Scientific finding and performance

In terms of mechanical properties, the authors strongly emphasized the effect of high density [2]rotaxane for tough materials. However, similar scientific finding using cyclic molecules, not crown ether derivatives though, already reported by other groups. [For example, Angew. Chem. Int. Ed. 2012, 51, 12504-12508, J. Am. Chem. Soc. 2017, 139, 42, 14825-14828, Chem. Rev. 2020, 120, 1, 310-433, etc. Furthermore, mechanical performance of the materials in this manuscript was very similar to the previously reported materials.

Point-by-Point Response to Reviewers' Comments

For Reviewer 1:

This revised manuscript can be published as it is now.

- We thank the reviewer for the recommendation of our revised manuscript for publication.

For Reviewer 2:

In the revised manuscript by Yan et al, they have collected the data for the reference polymer and addressed the temperature dependence of the MIMs. They also provided the clear evidence on the ionic interactions. I believe that they have made a better version according to the referees' comments. However, there is room for further improvement and I list my suggestions below for their references. I would like to support the publication on Nature Communication after minor revision.

- We thank the reviewer for the positive comments. According to the valuable suggestions, we have further improved our manuscript and the raised concerns have also been elaborately addressed as shown below.
1. Despite the authors have removed the discussion on the comparison between MIM-4 and other reported RCPs in the SI, they should also remove the related discussion in the main text and Figure 3e. Such discussion and presentation provide a clear misleading on the fact that the swelling ratio is highly dependent on the polymer backbone. Instead of putting other PCRs, the author should focus on MIN-1 to MIN-7 here if they want to highlight the mechanical performance.
 - We thank the reviewer for the kind suggestion. As suggested, we have removed the related figure and description in the main text to eliminate the misunderstanding. Meanwhile, we focus on the swellability and mechanical properties of MINs-1–6 to highlight the role of densely rotaxanated backbones on their mechanical performances, and corresponding discussion has also been revised in the main text on pages 6 and 7.
 2. I would suggest authors to put the temperature dependence results of the MIM-4 and the control data in the main figures and have a proper discussion, this is an important aspect for the MIMs based on H-G interactions.
 - We thank the reviewer for the kind reminder. As suggested, the result comparison of temperature sweep for the two samples was shown in Fig. I. The curves of storage modulus for the two samples could be roughly divided into three regions. The low-temperature regions were the glassy regimes with frozen networks, where the moduli of the two curves were close to each other due to their similar crosslinking density. In the intermediate regions, sharp decrease of the modulus was observed for both MIN-4 and control, which was caused by the glass transition of the networks. For the terminal regions with temperatures higher than 65 °C, significant difference for the two curves were observed. The curve of control exhibited an obvious plateau, whereas the modulus for the curve of

MIN-4 decreased slowly with the increase of temperatures, and its final value was obviously lower than that of control. It has been proved that host-guest recognition based on B21C7 and secondary ammonium salt could dissociate at temperatures higher than 70 °C (Yan, X. et al. *J. Am. Chem. Soc.* **142**, 2051–2058 (2020)). These phenomena might be related to the intramolecular motion of [2]rotaxane after dissociation of host-guest recognition, which can extend the network of MIN-4 and thus lead to a lower modulus. Basically, intramolecular motion of [2]rotaxane in control cannot affect the size of the network. Hence, similar phenomenon was not present in the curve of control. The figure has been added in Supplementary Information, and corresponding discussion has been added into the main text on page 8.

Fig. I (newly added Supplementary Fig. S22) DMA temperature ramps of MIN-4 and control from -50 to 100 °C with a heating rate of 5.0 °C/min.

3. I agree that the authors are trying to elucidate the integration and amplification effects on the mechanical bond in such polymeric systems, but the authors should indicate/emphasise what are the real merits arising from the densely rotaxanated backbones instead of putting a small amount of rotaxane in a cross-linked polymer.
 - We thank the reviewer for the valuable comment. To better highlight the differences and merits of the two systems, we have added detailed discussion in the section of mechanism study. In specific, a new paragraph of “Such a mechanism is largely different from that of the well-known rotaxane-crosslinked polymers [Takata, T. *et al. ACS Macro Lett.* **4**, 598–601 (2015); *Angew. Chem. Int. Ed.* **58**, 2765–2768 (2019)]. From the perspective of structure, the content of rotaxane units in the system is relatively low, and the main part of the network is still the conventional polymeric backbone. In this case, the rotaxane acts as a crucial functional unit which could modulate the stress distribution on conventional polymer chains. In a word, a small amount of rotaxane units is able to efficiently tune and improve the mechanical properties of conventional polymers, which represents a very smart and efficient design. For our MINs with densely rotaxanated backbones, rotaxane is the monomeric unit and main constitution of the network. The mechanical properties of MINs fully originate from integrating and amplifying the motion of rotaxanes. Therefore, the macroscopic mechanical properties of MINs are the direct reflection of microscopic motion of [2]rotaxanes. To sum up, rotaxane-crosslinked polymers and our MINs with densely rotaxanted backbones are two different systems, and each of them has its own working mechanisms and merits. For example, the preparation of rotaxane-crosslinked polymers is facile and they hold great potential in practical applications. However, the integration and amplification of stimuli-induced microscopic motion of rotaxanes

represent a unique mechanism in polymer physics which is also worth in-depth study from the fundamental point of view. Moreover, the integration and amplification of microscopic motion in MINs would result in superior actuation behaviors on which we will put more efforts in our future work.” has been added on pages 13 and 14.

4. The authors mentioned that the preparation of the deactivated rotaxane reference was failed. What are the conditions the authors used? Could authors try strong bases (such as DBU, NaH) to deprotonate the crown-ether ammonium salt instead of acetylation.

➤ We thank the reviewer for the kind suggestion. The preparation condition of the deactivated rotaxane was reference to the method reported by Takata *et al.* (*Chem. Eur. J.* **22**, 8759–8762 (2016)). In specific, the experimental condition was described as follows: In a round-bottom flask, [2]rotaxane **1** (50.0 mg, 0.0420 mmol), acetic anhydride (0.20 mL, 2.08 mmol) and triethylamine (0.61 mL, 4.15 mmol) in THF (5.0 mL) was stirred for 24 h. The solution was diluted with CH₂Cl₂ and washed with NaHCO₃ solution and water, dried over MgSO₄, filtered, evaporated in *vacuo*, and purified by flash column chromatography. The results of ¹H NMR showed that there was no change for the spectra of [2]rotaxane **1** before and after the experiment (Fig. IIa and b). Therefore, our attempt to deactivate the rotaxane was failed.

In order to tackle the problem, we have further performed new experiments using strong bases of DBU and NaH, according to the suggestion of the reviewer. The specific experimental conditions were as follows:

(1) In a round-bottom flask, [2]rotaxane **1** (50.0 mg, 0.0420 mmol) and 1,8-diazabicyclo[5.4.0]undec-7-ene (DBU, 31.5 mg, 0.210 mmol) in CH₃CN (3.0 mL) was stirred for 24 h. The reaction mixture was concentrated and purified by flash column chromatography (Qu, D.-H. et al. *Asian J. Org. Chem.* **7**, 902–905 (2018)).

(2) In a round-bottom flask, [2]rotaxane **1** (50.0 mg, 0.0420 mmol) and NaH (5.05 mg, 0.210 mmol) in CH₃CN (3.0 mL) was stirred for 24 h at 40 °C. The reaction mixture was cooled to room temperature, concentrated and then purified by flash column chromatography (Osakada, K. et al. *Macromolecules* **33**, 2315–2319 (2000)).

Unfortunately, the deactivation of the [2]rotaxane **1** was still not possible even using these two bases. As shown in Fig. II, the characteristic peaks of aromatic protons and the ethyleneoxy protons for [2]rotaxane **1** basically remained unchanged after the treatments with DBU (Fig. IIc) and NaH (Fig. IId), respectively, indicative of the unchanged chemical structure. We thank the reviewer again for this kind reminder. The suggestion is valuable and useful, and corresponding investigation will be conducted for our polymer system bearing DB24C8-based rotaxanes in the near future.

Fig. II Partial ^1H NMR spectra (CDCl_3 , 400 MHz, 293K) of [2]rotaxane **1** (a), and that after the treatments with triethylamine and acetic anhydride (b), DBU (c), and NaH (d), respectively.

For Reviewer 3:

The authors prepared [2]rotaxane-based polymeric materials with high mechanical properties. The authors experiments and discussions to evaluate amplification effect in molecular level were interesting and well-designed. However, we do not recommend accepting this manuscript in Nature Communication because of the reasons in below. It should be suitable more specified journal as the author studied well about their materials from various aspects.

- We thank the reviewer for his/her attention to our work. Based on the comments, we found that the main concerns of the reviewer concentrate on the novelty of our research, including the structural design, material performance, and the scientific finding. In fact, our work is significantly different from the cases mentioned by the reviewer, and represents a great step forward in mechanism study of mechanically interlocked polymers. In specific, the concerns raised by the reviewer have been well explained as shown below.

1. Materials design

The authors' molecular design was similar to a molecular design reported by Takata et al. [Angew. Chem. Int. Ed., 2019, 58, 2765-2768, DOI:10.1002/anie.201813439] and the recently published molecular design by authors [J. Am. Chem. Soc. 2022, 144, 872-882, DOI: 10.1021/jacs.1c10427].

- We thank the reviewer for his/her attention on our work. First of all, we would like to point out that the aim of our research is to reveal the integration and amplification mechanism of MINs. Therefore, whether the MINs are good model systems for mechanism study, which is highly important in our research. As for a proper model system, we think that it should meet the following requirements: (1) Integration and amplification effect depends on the accumulation of numerous microscopic mechanical motions, hence mechanical interlocked unit with high-density is a prerequisite for the structure of model MINs; (2) The structure of a model system should be simple and concise with interference factors as few as possible; (3) The model system should be general and universal, which could serve as a representative of one kind of mechanically interlocked polymers. Considering these

factors, it is obvious that our MINs are suitable model systems to study the integration and amplification mechanism. Benefitting from the proper structural design, the mechanism has been successfully elucidated in our research, which would act as a guiding strategy to design novel mechanically interlocked materials.

Actually, our research is also significantly different from the two works mentioned by the reviewers. In specific, the differences could be described as follows:

(1) For the work reported by Takata *et al.*, they investigated the role of mobile distance and the mobility of the rotaxane components on the mechanical properties of rotaxane-crosslinked polymers (RCPs). For the difference in terms of structure, the rotaxane units of RCPs act as discrete cross-links of polyolefin whose contents are low (≤ 0.5 mol%, as depicted in Fig. 1b) in the system, but the rotaxane moieties are the monomeric units in our MINs with high-density. As for the difference in working mechanism, a very few rotaxane cross-links in RCPs account for modulating the force distribution on the conventional polymeric chains through their sliding motion. Corresponding experiments have revealed that the mechanical properties of RCPs are much superior to those of covalently-cross-linked polymers, indicating the advantages of rotaxane cross-links to modulate the properties of conventional polymers. However, for our MINs, their macroscopic mechanical properties originate from the accumulation of numerous microscopic motions of the rotaxanes.

(2) For our previous work published in *JACS*, we coupled the vitrimer chemistry with mechanically interlocked structures to generate mechanically interlocked vitrimers (MIVs). By virtue of the vitrimer chemistry, we imparted reprocessability and chemical recyclability to the MINs, achieving the customization and sustainability of MINs. Though macroscopic mechanical properties of MIVs have also been investigated, it is intended to disclose that MIVs could possess mechanical properties and processibility simultaneously. Hence, the research target of current work is markedly different from the one published in *JACS*. Besides, the molecular design in these two works was also different. There are two kinds of dynamic interactions in the MIVs, namely, mechanical bond and dynamic covalent bond. Therefore, MIVs exhibit more complicated dynamic interactions, and are not suitable to be employed as the model system to reveal the integration and amplification mechanism.

2. Scientific finding and performance

In terms of mechanical properties, the authors strongly emphasized the effect of high density [2]rotaxane for tough materials. However, similar scientific finding using cyclic molecules, not crown ether derivatives though, already reported by other groups. [For example, *Angew. Chem. Int. Ed.* 2012, 51, 12504-12508, *J. Am. Chem. Soc.* 2017, 139, 42, 14825-14828, *Chem. Rev.* 2020, 120, 1, 310-433, etc. Furthermore, mechanical performance of the materials in this manuscript was very similar to the previously reported materials.

- We thank the reviewer for the comments. The examples the reviewer mentioned here inspire us a lot and we especially introduced the results by Prof. Giuseppone (*Angew. Chem. Int. Ed.* **51**, 12504–12508 (2012), ref. 26) in the Introduction part of our manuscript. In fact, the integration and amplification mechanism of mechanically interlocked polymers

is a significant scientific question in the field. Different from the methods or results reported in the mentioned literatures, we designed a new model system and developed a well-designed method to answer the question from our perspective.

For the scientific finding, Giuseppone and coworkers constructed a metallocsupramolecular poly[*c*2]daisy chain, wherein the scale change of the subunits could be amplified four orders of magnitude in solutions (Giuseppone et al. *Angew. Chem. Int. Ed.* **51**, 12504–12508 (2012)). Furthermore, they synchronized the movement of mechanical bonds at the molecular level to the macroscopic contraction and expansion of gel materials through acid-base stimulation (Giuseppone et al. *J. Am. Chem. Soc.* **139**, 14825–14828 (2017)). In these works, the molecular-level motion has been amplified into mesoscopic and macroscopic scales based on mechanically interlocked polymers, which laid an indelible foundation for our research. However, these studies focus on the scale change brought about by the integration mechanism in solution or in the gel state, and how the integrated motion affects the macroscopic mechanical properties of elastomeric mechanically interlocked materials is a pending challenge. In our work, MINs as model systems showcase macroscopic mechanical properties in the solid state originating from the amplification of integrated microscopic motion of the embedded [2]rotaxane motifs. In our work, we establish the structure–property relationship between cumulative microscopic motions and amplified macroscopic mechanical performance, which is a long-term pursuit since the first mechanically interlocked polymers were synthesized more than 40 years ago (Frisch, H. L. et al. *Polym. Eng. Sci.* **15**, 70–72 (1975); Wada, J. et al. *J. Polym. Sci. Polym. Ed.* **14**, 459–462 (1975)). In particular, we develop a continuous stress-relaxation approach to effectively elucidate the microscopic motion of [2]rotaxanes in bulk MINs, thereby providing a powerful yet universal tool to deeply understand the motion of mechanical bonds in the solid state.

For the mechanical performance, compared with the control whose wheels are nonslidable under applied force, our representative MIN showcases enhanced mechanical performance including Young's modulus (101 vs 1.68 MPa), toughness (47.3 vs 1.70 MJ/m³), damping capacity (95.4% vs 50.7%), and puncture energy (23.5 vs 1.09 mJ). We investigate the mechanical performances of the MINs with the aim of establishing their relationship with the force-induced microscopic motions of [2]rotaxanes. Such a goal has been achieved well in our work. As for the issue proposed by the reviewer that the mechanical performance of MINs in this manuscript was similar to our previous cases (Young's modulus: 101 (this work) vs 18.5 (our MIVs work) MPa; Toughness 47.3 (this work) vs 3.7 (our MIVs work) MJ/m³) (*J. Am. Chem. Soc.* **144**, 872–882 (2022)), it has its rationality in certain degree. Because though there are two kinds of dynamic interactions, mechanical properties of MIVs are mainly determined by the mechanical bonds, thus showing that the mechanical bonds hold great promise in modulating mechanical properties of polymers. However, the mechanism of integration and amplification has not been elucidated in that work, because the complicated system of MIVs is not suitable for corresponding study as we interpreted above.

Reviewers' Comments:

Reviewer #2:

Remarks to the Author:

The authors have addressed the questions I raised as far as they reasonably can. The manuscript is therefore suitable for publication.

Point-by-Point Response to Reviewers' Comments

For Reviewer 1:

This revised manuscript can be published as it is now.

- We thank the reviewer for the recommendation of our revised manuscript for publication.

For Reviewer 2:

In the revised manuscript by Yan et al, they have collected the data for the reference polymer and addressed the temperature dependence of the MIMs. They also provided the clear evidence on the ionic interactions. I believe that they have made a better version according to the referees' comments. However, there is room for further improvement and I list my suggestions below for their references. I would like to support the publication on Nature Communication after minor revision.

- We thank the reviewer for the positive comments. According to the valuable suggestions, we have further improved our manuscript and the raised concerns have also been elaborately addressed as shown below.
1. Despite the authors have removed the discussion on the comparison between MIM-4 and other reported RCPs in the SI, they should also remove the related discussion in the main text and Figure 3e. Such discussion and presentation provide a clear misleading on the fact that the swelling ratio is highly dependent on the polymer backbone. Instead of putting other PCRs, the author should focus on MIN-1 to MIN-7 here if they want to highlight the mechanical performance.
 - We thank the reviewer for the kind suggestion. As suggested, we have removed the related figure and description in the main text to eliminate the misunderstanding. Meanwhile, we focus on the swellability and mechanical properties of MINs-1–6 to highlight the role of densely rotaxanated backbones on their mechanical performances, and corresponding discussion has also been revised in the main text on pages 6 and 7.
 2. I would suggest authors to put the temperature dependence results of the MIM-4 and the control data in the main figures and have a proper discussion, this is an important aspect for the MIMs based on H-G interactions.
 - We thank the reviewer for the kind reminder. As suggested, the result comparison of temperature sweep for the two samples was shown in Fig. I. The curves of storage modulus for the two samples could be roughly divided into three regions. The low-temperature regions were the glassy regimes with frozen networks, where the moduli of the two curves were close to each other due to their similar crosslinking density. In the intermediate regions, sharp decrease of the modulus was observed for both MIN-4 and control, which was caused by the glass transition of the networks. For the terminal regions with temperatures higher than 65 °C, significant difference for the two curves were observed. The curve of control exhibited an obvious plateau, whereas the modulus for the curve of

MIN-4 decreased slowly with the increase of temperatures, and its final value was obviously lower than that of control. It has been proved that host–guest recognition based on B21C7 and secondary ammonium salt could dissociate at temperatures higher than 70 °C (Yan, X. et al. *J. Am. Chem. Soc.* **142**, 2051–2058 (2020)). These phenomena might be related to the intramolecular motion of [2]rotaxane after dissociation of host–guest recognition, which can extend the network of MIN-4 and thus lead to a lower modulus. Basically, intramolecular motion of [2]rotaxane in control cannot affect the size of the network. Hence, similar phenomenon was not present in the curve of control. The figure has been added in Supplementary Information, and corresponding discussion has been added into the main text on page 8.

Fig. I (newly added Supplementary Fig. S22) DMA temperature ramps of MIN-4 and control from –50 to 100 °C with a heating rate of 5.0 °C/min.

3. I agree that the authors are trying to elucidate the integration and amplification effects on the mechanical bond in such polymeric systems, but the authors should indicate/emphasise what are the real merits arising from the densely rotaxanated backbones instead of putting a small amount of rotaxane in a cross-linked polymer.
- We thank the reviewer for the valuable comment. To better highlight the differences and merits of the two systems, we have added detailed discussion in the section of mechanism study. In specific, a new paragraph of “Such a mechanism is largely different from that of the well-known rotaxane-crosslinked polymers [Takata, T. *et al. ACS Macro Lett.* **4**, 598–601 (2015); *Angew. Chem. Int. Ed.* **58**, 2765–2768 (2019)]. From the perspective of structure, the content of rotaxane units in the system is relatively low, and the main part of the network is still the conventional polymeric backbone. In this case, the rotaxane acts as a crucial functional unit which could modulate the stress distribution on conventional polymer chains. In a word, a small amount of rotaxane units is able to efficiently tune and improve the mechanical properties of conventional polymers, which represents a very smart and efficient design. For our MINs with densely rotaxanated backbones, rotaxane is the monomeric unit and main constitution of the network. The mechanical properties of MINs fully originate from integrating and amplifying the motion of rotaxanes. Therefore, the macroscopic mechanical properties of MINs are the direct reflection of microscopic motion of [2]rotaxanes. To sum up, rotaxane-crosslinked polymers and our MINs with densely rotaxanted backbones are two different systems, and each of them has its own working mechanisms and merits. For example, the preparation of rotaxane-crosslinked polymers is facile and they hold great potential in practical applications. However, the integration and amplification of stimuli-induced microscopic motion of rotaxanes

represent a unique mechanism in polymer physics which is also worth in-depth study from the fundamental point of view. Moreover, the integration and amplification of microscopic motion in MINs would result in superior actuation behaviors on which we will put more efforts in our future work.” has been added on pages 13 and 14.

4. The authors mentioned that the preparation of the deactivated rotaxane reference was failed. What are the conditions the authors used? Could authors try strong bases (such as DBU, NaH) to deprotonate the crown-ether ammonium salt instead of acetylation.

➤ We thank the reviewer for the kind suggestion. The preparation condition of the deactivated rotaxane was reference to the method reported by Takata *et al.* (*Chem. Eur. J.* **22**, 8759–8762 (2016)). In specific, the experimental condition was described as follows: In a round-bottom flask, [2]rotaxane **1** (50.0 mg, 0.0420 mmol), acetic anhydride (0.20 mL, 2.08 mmol) and triethylamine (0.61 mL, 4.15 mmol) in THF (5.0 mL) was stirred for 24 h. The solution was diluted with CH₂Cl₂ and washed with NaHCO₃ solution and water, dried over MgSO₄, filtered, evaporated in *vacuo*, and purified by flash column chromatography. The results of ¹H NMR showed that there was no change for the spectra of [2]rotaxane **1** before and after the experiment (Fig. IIa and b). Therefore, our attempt to deactivate the rotaxane was failed.

In order to tackle the problem, we have further performed new experiments using strong bases of DBU and NaH, according to the suggestion of the reviewer. The specific experimental conditions were as follows:

(1) In a round-bottom flask, [2]rotaxane **1** (50.0 mg, 0.0420 mmol) and 1,8-diazabicyclo[5.4.0]undec-7-ene (DBU, 31.5 mg, 0.210 mmol) in CH₃CN (3.0 mL) was stirred for 24 h. The reaction mixture was concentrated and purified by flash column chromatography (Qu, D.-H. et al. *Asian J. Org. Chem.* **7**, 902–905 (2018)).

(2) In a round-bottom flask, [2]rotaxane **1** (50.0 mg, 0.0420 mmol) and NaH (5.05 mg, 0.210 mmol) in CH₃CN (3.0 mL) was stirred for 24 h at 40 °C. The reaction mixture was cooled to room temperature, concentrated and then purified by flash column chromatography (Osakada, K. et al. *Macromolecules* **33**, 2315–2319 (2000)).

Unfortunately, the deactivation of the [2]rotaxane **1** was still not possible even using these two bases. As shown in Fig. II, the characteristic peaks of aromatic protons and the ethyleneoxy protons for [2]rotaxane **1** basically remained unchanged after the treatments with DBU (Fig. IIc) and NaH (Fig. IId), respectively, indicative of the unchanged chemical structure. We thank the reviewer again for this kind reminder. The suggestion is valuable and useful, and corresponding investigation will be conducted for our polymer system bearing DB24C8-based rotaxanes in the near future.

Fig. II Partial ^1H NMR spectra (CDCl_3 , 400 MHz, 293K) of [2]rotaxane **1** (a), and that after the treatments with triethylamine and acetic anhydride (b), DBU (c), and NaH (d), respectively.

For Reviewer 3:

The authors prepared [2]rotaxane-based polymeric materials with high mechanical properties. The authors experiments and discussions to evaluate amplification effect in molecular level were interesting and well-designed. However, we do not recommend accepting this manuscript in Nature Communication because of the reasons in below. It should be suitable more specified journal as the author studied well about their materials from various aspects.

- We thank the reviewer for his/her attention to our work. Based on the comments, we found that the main concerns of the reviewer concentrate on the novelty of our research, including the structural design, material performance, and the scientific finding. In fact, our work is significantly different from the cases mentioned by the reviewer, and represents a great step forward in mechanism study of mechanically interlocked polymers. In specific, the concerns raised by the reviewer have been well explained as shown below.

1. Materials design

The authors' molecular design was similar to a molecular design reported by Takata et al. [Angew. Chem. Int. Ed., 2019, 58, 2765-2768, DOI:10.1002/anie.201813439] and the recently published molecular design by authors [J. Am. Chem. Soc. 2022, 144, 872-882, DOI: 10.1021/jacs.1c10427].

- We thank the reviewer for his/her attention on our work. First of all, we would like to point out that the aim of our research is to reveal the integration and amplification mechanism of MINs. Therefore, whether the MINs are good model systems for mechanism study, which is highly important in our research. As for a proper model system, we think that it should meet the following requirements: (1) Integration and amplification effect depends on the accumulation of numerous microscopic mechanical motions, hence mechanical interlocked unit with high-density is a prerequisite for the structure of model MINs; (2) The structure of a model system should be simple and concise with interference factors as few as possible; (3) The model system should be general and universal, which could serve as a representative of one kind of mechanically interlocked polymers. Considering these

factors, it is obvious that our MINs are suitable model systems to study the integration and amplification mechanism. Benefitting from the proper structural design, the mechanism has been successfully elucidated in our research, which would act as a guiding strategy to design novel mechanically interlocked materials.

Actually, our research is also significantly different from the two works mentioned by the reviewers. In specific, the differences could be described as follows:

(1) For the work reported by Takata *et al.*, they investigated the role of mobile distance and the mobility of the rotaxane components on the mechanical properties of rotaxane-crosslinked polymers (RCPs). For the difference in terms of structure, the rotaxane units of RCPs act as discrete cross-links of polyolefin whose contents are low (≤ 0.5 mol%, as depicted in Fig. 1b) in the system, but the rotaxane moieties are the monomeric units in our MINs with high-density. As for the difference in working mechanism, a very few rotaxane cross-links in RCPs account for modulating the force distribution on the conventional polymeric chains through their sliding motion. Corresponding experiments have revealed that the mechanical properties of RCPs are much superior to those of covalently-cross-linked polymers, indicating the advantages of rotaxane cross-links to modulate the properties of conventional polymers. However, for our MINs, their macroscopic mechanical properties originate from the accumulation of numerous microscopic motions of the rotaxanes.

(2) For our previous work published in *JACS*, we coupled the vitrimer chemistry with mechanically interlocked structures to generate mechanically interlocked vitrimers (MIVs). By virtue of the vitrimer chemistry, we imparted reprocessability and chemical recyclability to the MINs, achieving the customization and sustainability of MINs. Though macroscopic mechanical properties of MIVs have also been investigated, it is intended to disclose that MIVs could possess mechanical properties and processibility simultaneously. Hence, the research target of current work is markedly different from the one published in *JACS*. Besides, the molecular design in these two works was also different. There are two kinds of dynamic interactions in the MIVs, namely, mechanical bond and dynamic covalent bond. Therefore, MIVs exhibit more complicated dynamic interactions, and are not suitable to be employed as the model system to reveal the integration and amplification mechanism.

2. Scientific finding and performance

In terms of mechanical properties, the authors strongly emphasized the effect of high density [2]rotaxane for tough materials. However, similar scientific finding using cyclic molecules, not crown ether derivatives though, already reported by other groups. [For example, *Angew. Chem. Int. Ed.* 2012, 51, 12504-12508, *J. Am. Chem. Soc.* 2017, 139, 42, 14825-14828, *Chem. Rev.* 2020, 120, 1, 310-433, etc. Furthermore, mechanical performance of the materials in this manuscript was very similar to the previously reported materials.

- We thank the reviewer for the comments. The examples the reviewer mentioned here inspire us a lot and we especially introduced the results by Prof. Giuseppone (*Angew. Chem. Int. Ed.* **51**, 12504–12508 (2012), ref. 26) in the Introduction part of our manuscript. In fact, the integration and amplification mechanism of mechanically interlocked polymers

is a significant scientific question in the field. Different from the methods or results reported in the mentioned literatures, we designed a new model system and developed a well-designed method to answer the question from our perspective.

For the scientific finding, Giuseppone and coworkers constructed a metallocsupramolecular poly[*c*2]daisy chain, wherein the scale change of the subunits could be amplified four orders of magnitude in solutions (Giuseppone et al. *Angew. Chem. Int. Ed.* **51**, 12504–12508 (2012)). Furthermore, they synchronized the movement of mechanical bonds at the molecular level to the macroscopic contraction and expansion of gel materials through acid-base stimulation (Giuseppone et al. *J. Am. Chem. Soc.* **139**, 14825–14828 (2017)). In these works, the molecular-level motion has been amplified into mesoscopic and macroscopic scales based on mechanically interlocked polymers, which laid an indelible foundation for our research. However, these studies focus on the scale change brought about by the integration mechanism in solution or in the gel state, and how the integrated motion affects the macroscopic mechanical properties of elastomeric mechanically interlocked materials is a pending challenge. In our work, MINs as model systems showcase macroscopic mechanical properties in the solid state originating from the amplification of integrated microscopic motion of the embedded [2]rotaxane motifs. In our work, we establish the structure–property relationship between cumulative microscopic motions and amplified macroscopic mechanical performance, which is a long-term pursuit since the first mechanically interlocked polymers were synthesized more than 40 years ago (Frisch, H. L. et al. *Polym. Eng. Sci.* **15**, 70–72 (1975); Wada, J. et al. *J. Polym. Sci. Polym. Ed.* **14**, 459–462 (1975)). In particular, we develop a continuous stress-relaxation approach to effectively elucidate the microscopic motion of [2]rotaxanes in bulk MINs, thereby providing a powerful yet universal tool to deeply understand the motion of mechanical bonds in the solid state.

For the mechanical performance, compared with the control whose wheels are nonslidable under applied force, our representative MIN showcases enhanced mechanical performance including Young's modulus (101 vs 1.68 MPa), toughness (47.3 vs 1.70 MJ/m³), damping capacity (95.4% vs 50.7%), and puncture energy (23.5 vs 1.09 mJ). We investigate the mechanical performances of the MINs with the aim of establishing their relationship with the force-induced microscopic motions of [2]rotaxanes. Such a goal has been achieved well in our work. As for the issue proposed by the reviewer that the mechanical performance of MINs in this manuscript was similar to our previous cases (Young's modulus: 101 (this work) vs 18.5 (our MIVs work) MPa; Toughness 47.3 (this work) vs 3.7 (our MIVs work) MJ/m³) (*J. Am. Chem. Soc.* **144**, 872–882 (2022)), it has its rationality in certain degree. Because though there are two kinds of dynamic interactions, mechanical properties of MIVs are mainly determined by the mechanical bonds, thus showing that the mechanical bonds hold great promise in modulating mechanical properties of polymers. However, the mechanism of integration and amplification has not been elucidated in that work, because the complicated system of MIVs is not suitable for corresponding study as we interpreted above.

REVIEWERS' COMMENTS

Reviewer #2 (Remarks to the Author):

The authors have addressed the questions I raised as far as they reasonably can. The manuscript is therefore suitable for publication.

Response to Reviewers' Comments

Reviewer #1

Comment

The authors have addressed the questions I raised as far as they reasonably can. The manuscript is therefore suitable for publication.

Response: We thank the reviewer for their positive assessment of the revised manuscript and for supporting its publication.